# Forward and Backward Recalling Sequences in Spatial and Verbal Memory Tasks: What Do We Measure?

**DOI:** 10.3390/e25050813

**Published:** 2023-05-18

**Authors:** Jeanette Melin, Laura Göschel, Peter Hagell, Albert Westergren, Agnes Flöel, Leslie Pendrill

**Affiliations:** 1Research Institutes of Sweden (RISE), Division Safety and Transport, Department of Measurement Science and Technology, 41258 Gothenburg, Sweden; leslie.pendrill@ri.se; 2Department of Leadership, Demand and Control, Swedish Defence University, 65340 Karlstad, Sweden; 3Department of Neurology, Charité—Universitätsmedizin Berlin, Corporate Member of Freie Universität Berlin and Humboldt-Universität zu Berlin, Charitéplatz 1, 10117 Berlin, Germany; laura.goeschel@charite.de; 4NCRC—Neuroscience Clinical Research Center, Charité—Universitätsmedizin Berlin, Corporate Member of Freie Universität Berlin and Humboldt-Universität zu Berlin, Charitéplatz 1, 10117 Berlin, Germany; 5The PRO-CARE Group, Faculty of Health Sciences, Kristianstad University, 29188 Kristianstad, Sweden; peter.hagell@hkr.se (P.H.); albert.westergren@hkr.se (A.W.); 6The Research Platform for Collaboration for Health, Faculty of Health Sciences, Kristianstad University, 29188 Kristianstad, Sweden; 7Department of Neurology, University Medicine Greifswald, 17475 Greifswald, Germany; agnes.floeel@med.uni-greifswald.de; 8German Center for Neurodegenerative Diseases (DZNE), Standort, 17475 Greifswald, Germany

**Keywords:** cognition, visuo-spatial memory, verbal memory, metrology, neuropsychology, neuropsychological assessments, neurodegenerative diseases, cognitive neuroscience, Rasch

## Abstract

There are different views in the literature about the number and inter-relationships of cognitive domains (such as memory and executive function) and a lack of understanding of the cognitive processes underlying these domains. In previous publications, we demonstrated a methodology for formulating and testing cognitive constructs for visuo-spatial and verbal recall tasks, particularly for working memory task difficulty where entropy is found to play a major role. In the present paper, we applied those insights to a new set of such memory tasks, namely, backward recalling block tapping and digit sequences. Once again, we saw clear and strong entropy-based construct specification equations (CSEs) for task difficulty. In fact, the entropy contributions in the CSEs for the different tasks were of similar magnitudes (within the measurement uncertainties), which may indicate a shared factor in what is being measured with both forward and backward sequences, as well as visuo-spatial and verbal memory recalling tasks more generally. On the other hand, the analyses of dimensionality and the larger measurement uncertainties in the CSEs for the backward sequences suggest that caution is needed when attempting to unify a single unidimensional construct based on forward and backward sequences with visuo-spatial and verbal memory tasks.

## 1. Introduction

In order to evaluate individuals’ cognitive abilities and cognitive processes in general, several neuropsychological tests have been developed to measure different cognitive domains. Typically, individual neuropsychological tests are designed to measure one or more discrete abilities. There are, however, different understandings in the literature about the number and inter-relationships of cognitive domains (such as memory and executive function) [1,2]. As highlighted by many, memory is the most multidimensional cognitive domain [3,4]. Among others, it includes working memory, which can be defined in several ways [4] but has commonly been referred to as the ability to temporarily maintain (verbal and nonverbal) information in consciousness for adaptive use in information processing [1]. Executive functions, for their part, include several actions and processes [2] to real-world adaptive tasks [1].

Our previous studies of neuropsychological tests in the NeuroMET project have focused on the recall of forward sequences of blocks or digits (i.e., working memory) [5,6,7] or freely recalling words (i.e., declarative memory) [8,9,10]. The recall of forward sequences of blocks and digits is typically regarded as engaging the visuo-spatial (blocks) and verbal (digits) maintenance working memory, respectively [1]. Backward recalling sequences, for their part, may be used to measure both visuo-spatial and verbal manipulation working memory, as well as executive functions [11,12,13]. However, previous studies have not yet been able to understand the cognitive processes underlying working memory, especially with regard to differences in forward or backward recalling sequences [14].

The Corsi Block Test (CBT) [15] and Digit Span Test (DST) [16] are two tests, largely free of cultural and language effects, that are commonly used in neuropsychological tests. Previous work has shown differences between visuo-spatial and verbal memory tasks [17,18], and forward and backward sequences seem to have different effects on visuo-spatial or verbal memory tasks [11,12,13,18]. It should be noted that visual sequences cannot form single coherent objects in the same way as verbal sequences do [19] and that digits are sampled from a smaller pool than other verbal stimuli such as letters or words [20]. Furthermore, despite decades of research, less is known about the tests with backward sequences compared with the forward sequences [21]. The “visuo-spatial hypothesis” has been accepted for many years [22,23], according to which backward recall, for instance, relies in part on visuo-spatial processes where the test person creates a mental list. More recent research by Guitard et al. [21] has challenged a too strict view of the visuo-spatial processes in backward recall. Specifically, they propose that phonological encoding is needed in forward and backward recalling sequences [21], and which test persons may rely more on when anticipating backward recalls [24,25,26].

The CBT and DST are two tests which are structured similarly, where the test person is first asked to recall sequences of increasing difficulty (which depends on sequence structure, such as length and number of reversals) of either taps on a board of blocks or spoken digits in a nominal “forward” direction. This is followed by asking the test person to inversely recall the same sequences of either taps or spoken digits but “backwards” instead. For either test, if the test person recalls a sequence correctly, the observed, “passed” response is scored 1, while an incorrect, “failed” response is scored 0. Traditionally, a sum of passes is then counted as a ‘measure’ of the test person’s ability.

Passes or fails in responses to memory tests are just observations, lying at best on an ordinal response scale, and simply counting the number of passes cannot be regarded as measurement, since the latter must be at least on an interval scale [7,27,28]. Ordinal observations (such as responses to memory tests) are subject to possible scale non-linearity, which, unless evaluated and corrected for, will lead to uncertainties and decision risks. Responses also depend on both task difficulty and person ability, and the same score can be obtained with an easy task performed by a less able person as well as for a difficult task performed by a more able person. To date, unfortunately, few analyses of memory tests have accounted for ordinality or addressed separability between item and person attributes. This has hindered the application of methods for metrological quality assurance to ensure traceability and declare measurement uncertainties in memory tests. We have previously pointed this out and argued for the following [29]: (i) applying a measurement system approach where the human responder acts as an instrument [30]; (ii) exploiting the unique properties of item response theory (IRT) models—particularly the Rasch model [31]– to compensate for ordinality and to establish metrological references; and (iii) formulating construct specification equations (CSE) to provide valid explanations of task difficulty and person ability [6,7,9,31,32].

CSEs were first introduced in the early 1980′s by Stenner and colleagues [31,32] and are frequently promoted as a means of ensuring validity [33,34,35,36], but the development of such construct theories has been slow in healthcare [6,36]. The formulation of CSEs, guided by an understanding of what is causing variation in a set of items, i.e., task difficulty, has been encouraged as a means of validating construct theories [32]. CSEs can provide a specific, causal, and rigorous mathematical conceptualization of working memory constructs (such as task difficulties [6]) and, in turn, support claims of the highest level of validity [34]. A set of items is expected to be used to measure one single person attribute, and, when there are equivalent CSEs for visuo-spatial and verbal memory tasks, they can validly be combined, thereby improving the reliability in the measurement of person ability [29]. On the contrary, if the CSE does not support unidimensionality between different sets of items, the validity in a higher ordered measurement is challenged [31,33], e.g., when combining visuo-spatial and verbal memory tasks or forward and backward sequences. For instance, in the present case, if the CSE cannot coherently and equally explain the variation in task difficulties from forward and backward sequences for both visuo-spatial and verbal memory tests, that may be an indication of a multidimensional memory domain.

Rasch modelling, according to its principle of specific objectivity, provides estimates of the level of difficulty of each test sequence across the whole cohort of test persons and separately estimates of the level of ability of each test person irrespective of test sequence [37]. The Rasch model is a probabilistic measurement model where a person who has better memory ability will be more likely to score higher on a difficult item than a person who has lower memory ability, and, conversely, it is more likely that more persons score high on an easy item and fewer persons score high on difficult items. By using these separate estimates of task difficulty in working memory tests in our previous publications, we have demonstrated both a methodology—CSE—for testing the validity of our understanding of constructs such as working memory task difficulty, including the particular role played by entropy when explaining working memory task difficulty [6,8]. Specifically, entropy is a measure of order—the higher the order, the lower the entropy, and a more ordered task will generally be easier to perform—which has been demonstrated experimentally [6,38]. Together with entropy, reversals and average distance (further described in the method section) have provided strong explanatory models for working memory task difficulty in CBT and DST forward sequences [6]. Moreover, the equal-entropy-based CSEs indicate the equivalence of different set items, which, together with the conceptual understanding and design of those working memory tests, is a key to understanding what is being measured.

In the present paper, we applied our previous insights to a new set of working memory tasks, namely, backward recalling block tapping and digit sequences, to provide a better understanding of what is being measured with CBT and DST backward. Firstly, we explore if and how well memory task difficulty can be explained in backward recalling sequences based on the same set of explanatory variables as for the forward sequences, as well as compare this between the visuo-spatial and verbal memory tests. Secondly, we aim to compare traditional Rasch-model-based analyses of dimensionality with CSE for memory task difficulty between forward and backward sequences, as well as between visuo-spatial and verbal memory tests.

## 2. Materials and Methods

### 2.1. Subjects

The data for this study stemmed from 332 individual assessments of the project NeuroMET [39] comprising 65 observations of patients with Alzheimer’s dementia (AD), 55 observations of patients with mild cognitive impairment (MCI), 102 observations of participants with subjective cognitive decline (SCD) and 110 observations of healthy controls (HC). The mean age of the participants was 72 years (SD 7, range 53–88 years). A more comprehensive report on recruitment of subjects and testing can be found elsewhere [29]. The number of observations used here depended on missing data for certain neuropsychological tests or missing data on biomarkers and might, therefore, not be equal to the number of observations in previous work. Nevertheless, the study population generally overlaps, and recruitment and testing procedures were identical.

### 2.2. Data Analysis

As mentioned in the Introduction, for both CBT and DST, the raw response scores (a correctly recalled sequence is scored 1; an incorrect response or missing due to the stopping rule is scored 0) were transformed with the dichotomous Rasch model [37] using restitution to provide separate and linear measures for individual task difficulties and person abilities on an interval scale. This restitution process used the RUMM2030Plus software (www.rummlab.com.au, accessed on 17 May 2023).

In order to explain task difficulty, the same kinds of explanatory variables for task difficulty, namely, *entropy*, *reversals* and *average distance*, as previously identified for CBT and DST forward sequences [6] (see Introduction), were also applied here to the corresponding backward sequences. A detailed presentation of those explanatory variables can be found in our previous publications [6,38], but, in short, the basic idea—that *entropy* refers to the fact that more ordered sequences will be easier to recall than less ordered sequences—as given in the Introduction was implemented here, specifically, by expecting that memory task difficulty, *δ*, would increase in proportion to ln(*G_j_*!), where *G* is the number of symbols encountered in a message and applies to sequences where there are no repeated symbols in the sequence. With repeats, memory task difficulty, *δ*, would be expected to decrease, as given by the following equation $\delta =Entropy=K\cdot \left[ln\left(G!\right)-\overset{M}{\underset{j=1}{\sum }}ln\left(N_{j}!\right)\right]$, where *N* is the number of repeats in the sequence of symbols of *M* different types (blocks or digits), and *K* is a normalization constant according to Brillouin [40]. *Reversal* corresponds to a changed direction in the sequence, i.e., changing from clockwise to anti-clockwise (or the other way around) in CBT, and from counting forwards to counting backwards (or the other way around) in DST. *Average distance* for CBT is the sum of distance in cm divided by the number of taps, and, for DST, it is the sum of the numerical distance between digits divided by the number of digits.

The explanatory variables observed may not be the principal components of variation in task difficulty, and state-of-the-art formulation of CSEs, therefore, includes three steps of a principal component regression [7,38]:i.A principal component analysis (PCA) amongst the set of explanatory variables, ***X_k_***;ii.A linear regression of the empirical task difficulty values *δ_j_* against ***X′*** in terms of the principal components, ***P***;iii.A conversion back from principal components to the explanatory variables, ***X_k_***.

Analyses of dimensionality based on the Rasch model [37] were conducted according to three approaches [41,42,43,44,45] using four subsets of data (CBT forward and backward sequences combined, DST forward and backward sequences combined, forward sequences from the CBT and DST combined, and backward sequences from the CBT and DST combined). First, PCAs of item residuals (not to be mixed with the PCA in the formulation of the CSE [8]) were conducted, and eigenvalues of the first and second principal components (PC1 and PC2, respectively) were compared. A large eigenvalue for PC1 (relative to PC2) may suggest multidimensionality [41]. Residuals are the standardized difference between observed and expected (Rasch model predicted probabilities) item responses or task outcomes. Since the analysis involves residuals representing what is left following the measurement process, the main dimension in the data has already been accounted for by the Rasch model. Therefore, under unidimensionality, no meaningful associations are expected among residuals, and PCA of residuals essentially assesses the extent to which additional dimensions may have influenced item responses. That is, low eigenvalues that decrease smoothly across PCs support unidimensionality [41,45]. Secondly, since the CBT and DST consist of forward and backward sequences that may represent different underpinning constructs, we constructed subtests (i.e., treated individual sequences of the same kind as a single polytomous item in the analyses) to identify potential multidimensionality in the data. Local independence is an assumption in measurement and may be violated in two principal ways: either if a secondary trait influences responses (i.e., multidimensionality, or local trait dependence) or if the outcome of one sequence (or response to one item) is influenced by the outcome of (or response to) another (i.e., local response dependence) [46]. Such dependencies may be accounted for by merging dependent items into subtests that are treated as single items in the analysis, and the outcomes of such analyses (relative to analyses that do not account for potential dependency) can be used to test for dependency. More specifically, if the reliability estimate (coefficient alpha) from a subtest analysis is lower than that from the first analysis (where all sequences are treated as individual items), multidimensionality is suggested. Further information is gained from the additional parameters *c* (variance that is unique to the subtests), *r* (the latent correlation between the subscales, corrected for attenuation due to measurement error), and *A* (the non-error variance common to all subtests), which will return a low value for *c* but high values for *A* and *r* when data are approximately unidimensional. Finally, unidimensionality was tested by comparing person locations (measures) based on items hypothesized to represent different dimensions (e.g., forward and backward sequences) using the independent *t*-test approach [41,44,45]. That is, since the Rasch model yields both locations and associated standard errors of each person in the analysis, the locations from the two sets of sequences were compared for each individual. If the proportion (or its 95% CI) of persons with significantly different location estimates is less than 5%, unidimensionality is considered supported. All dimensionality analyses were conducted using the RUMM2030Plus software (www.rummlab.com.au, accessed on 17 May 2023).

## 3. Results

The results section is divided into two parts: first, the CSE analyses are presented, and, secondly, traditional Rasch-model-based analyses of dimensionality are presented.

### 3.1. Construct Specification Equations

Equations (1) and (2) provide the CSEs obtained for the CBT forward and backward sequences (uncertainties in parentheses, coverage factor, *k* = 2):(1)$$zR_{j,\;CBT\_forward}=-6\;\left(2\right)+1.3\;\left(6\right)\times Entropy_{j}+0.1\;\left(1.1\right)\times Reversals_{j}+0.03\;\left(15\right)\times AveDist_{j}$$
(2)$$zR_{j,\;CBT\_backward}=-8\;\left(3\right)+1.5\;\left(1.0\right)\times Entropy_{j}-0.6\;\left(2.0\right)\times Reversals_{j}+0.2\;\left(2\right)\times AveDist_{j}$$

Equations (3) and (4) provide the CSEs obtained for DST forward and backward sequences:(3)$$zR_{j,\;DST\_forward}=-9\;\left(2\right)+1.2\;\left(4\right)\times Entropy_{j}+0.1\;\left(6\right)\times Reversals_{j}+0.04\;\left(37\right)\times AveDist_{j}$$
(4)$$zR_{j,\;DST\_backward}=-8\;\left(6\right)+1.3\;\left(1.7\right)\times Entropy_{j}-0.1\;\left(2.9\right)\times Reversals_{j}+0.8\;\left(7\right)\times AveDist_{j}$$

Firstly, as expected, the CSEs for the forward sequences corresponded well with our previous results [6,38]. Secondly, in all four CSEs, *entropy* was the only explanatory variable with measurement uncertainties smaller than the β-coefficient, thereby making it the dominating explanatory variable, which is what we could expect based on our previous work [6,38]. However, there were larger measurement uncertainties for the *entropy* β-coefficients in the CSEs for the backward sequences (Equations (2) and (4)) compared with the forward sequences (Equations (1) and (3)). There was also some more variation, e.g., the sign of the β-coefficient for *reversals* was changed from positive to negative for the CBT sequences (Equations (1) and (2)), and there were also larger measurement uncertainties for the β-coefficients for both *reversals* and *average distance* in the CSEs for the backward sequences (Equations (2) and (4)) compared with the forward sequences (Equations (1) and (3)).

Two further CSEs were derived based on items for forward and backward sequences in the CBT and on items for forward and backward sequences in the DST. This yielded the following CSEs:(5)$$zR_{j,\;CBT}=-7\;\left(1\right)+1.4\;\left(5\right)\times Entropy_{j}-0.3\;\left(9\right)\times Reversals_{j}+0.1\;\left(1\right)\times AveDist_{j}$$
(6)$$zR_{j,\;DST}=-8\;\left(5\right)+1.3\;\left(1.2\right)\times Entropy_{j}-0.3\;\left(2.1\right)\times Reversals_{j}+0.6\;\left(7\right)\times AveDist_{j}$$

As for the analyses of forward and backward sequences individually, when combined for block recalling sequences (i.e., the CBT) or digit recalling sequences (i.e., the DST), *entropy* continued to be the dominating explanatory variable for task difficulty, i.e., the uncertainties were smaller than the β-coefficients. For all explanatory variables, however, the uncertainties for the β-coefficients were particularly high for the digit recalling sequences (Equation (6)), while the β-coefficient for *entropy* decreased when CBT forward and backward sequences were combined (Equation (5)) compared to individually (Equations (1) and (2)).

Furthermore, CSEs based on a combination of items for forward sequences from CBT and DST, as well as items for backward sequences from CBT and DST, were derived:(7)$$zR_{j,forward}=-8\left(1\right)+1.3\left(4\right)\times Entropy_{j}-0.4\left(4\right)\times Reversals_{j}+0.3\left(1\right)\times AveDist_{j}$$
(8)$$zR_{j,backward}=-6\left(1\right)+1.1\left(3\right)\times Entropy_{j}-0.6\left(1.1\right)\times Reversals_{j}+0.1\left(2\right)\times AveDist_{j}$$

In these CSEs, *entropy* was again found to be dominating. Here, it should be noticed that *average distance* was also significant (i.e., measurement uncertainties for the β-coefficient not overlapping zero) and *reversals* were close to significant for the forward sequences (Equation (7)), but measurement uncertainties remained large for the backward sequences (Equation (8)).

Based on the CSEs, quasi-theoretical measures of memory task difficulty could be estimated and from all eight CSEs; those estimates were highly correlated with the empirical values, i.e., measures of task difficulties, *δ*, from the Rasch analysis (Pearson correlation coefficients ranged from 0.87 to 0.99). The lowest correlation was seen for the digit recalling sequences (Equation (6)), which was also the CSE with the largest measurement uncertainties in the β-coefficients. On the other hand, the strongest correlations were seen for the forward sequences individually (Equations (1) and (3)).

### 3.2. Dimensionality Analysis

The results are reported in Table 1. The PCA of forward and backward sequences combined yielded PC1 and PC2 eigenvalues (Section 2.2) of 1.88 and 1.7, respectively, for the CBT, with a PC1/PC2 ratio of 1.11. Equivalent results for the DST showed a somewhat higher PC1 eigenvalue and a PC1/PC2 ratio of 1.38. In both instances, forward and backward sequences tended to load in different directions for the PC1. When analyzing all forward sequences together and all backward sequences together, the PC1 eigenvalues increased to 2.34 (forward sequences) and 2.65 (backward sequences), both with a PC1/PC2 ratio of 1.44. In both instances, CBT and DST sequences tended to load in different directions for the PC1.

In all instances, coefficient alpha values decreased following the creation of subtests as compared to analyses based on individual sequences (Table 1). The decreases were more pronounced in analyses of combined CBT and DST forward and backward sequences (−0.23 and −0.21, respectively) than in analyses of combined forward and backward sequences of the CBT and DST (−0.06 and −0.07, respectively). In keeping with these observations, the values of *r* and *A* were relatively high for the combined forward and backward sequences of the CBT and DST (*r* was 0.85 and *A* was 0.92 for the CBT; *r* was 0.83 and *A* was 0.91 for the DST), whereas *c* was lower (0.42 and 0.46, respectively). In contrast, *r* and *A* were more modest for the combined CBT and DST forward (*r* was 0.54; *A* was 0.71) and backward (*r* was 0.57; *A* was 0.84) sequences, while the values of *c* were higher (0.93 and 0.87, respectively).

A similar pattern was found when using the independent t-test approach (Table 1). That is, there was some, but relatively weak evidence against unidimensionality when comparing person locations based on forward vs. backward sequences in the CBT, with significant different locations for 8.43% (95% CI, 5.86–11.96%) of the sample. Somewhat stronger evidence against unidimensionality was found from the equivalent analysis of the DST, where 13.55% (10.26–17.68%) of the sample had significantly different locations from the two types of sequences. When comparing forward sequences from the CBT and DST, there were clearer indications of multidimensionality, with 20.54% (16.53–25.23%) of the sample having significantly different locations. Very similar results were found from comparisons of the two backward sequences (Table 1). In all instances, the results suggested unidimensionality when the same analyses were conducted based on subtests (significantly different locations for 3.34–6.71% of the sample).

## 4. Discussion and Conclusions

This study has provided new insights into working memory and executive functions by means of a new approach to studying recall tests. Specifically, we have successfully demonstrated experimentally how entropy can be used to explain task difficulty in different combinations of recalling items, thus complementing our earlier work on forward sequences (see Introduction) with the corresponding backward sequences studied here. The very similar entropy contributions in the CSEs—within the measurement uncertainties—may indicate a shared dimension in what is being measured in both forward and backward sequences and visuo-spatial and verbal memory recalling tasks. Despite the fact that backward recalling sequences may be used to measure both visuo-spatial and verbal manipulation working memory as well as executive functions [11,12,13], it is often considered that backward sequences are dominated by executive functions. The present work has emphasized the interaction between the different cognitive dimensions. Previous studies claimed that forward sequences are different from backward sequences [11,12,13] or claimed a dissociation between visuo-spatial and verbal memory tasks [17,18]. We could not confirm these results based on the CSEs when considering our relatively large measurement uncertainties, (e.g., the entropy β-coefficients have overlapping ranges), Thus, our findings can be related to the on-going debate about whether working memory is an ability in itself or rather has to do with experiences relevant to the specific demands of the task [20,47].

Multidimensionality always exists to some extent [48,49], as was largely corroborated by our results from dimensionality analyses based on the Rasch model. That is, strictly speaking, there were signs of multidimensionality in all three analytical approaches of all four subsets of data. However, the dimensionality property is not an absolute but a relative matter of degree, and overreliance on statistical thresholds may be dangerous [44,50,51,52]. With this in mind, the observations reported here suggest that forward and backward sequences within the analyzed tests (i.e., visuo-spatial and verbal working memory, CBT and DST, respectively) appear to represent related but non-identical constructs. On the other hand, the different types of forward and backward sequences (i.e., maintenance or manipulation working memory, CBT and DST, respectively) appear less related and more likely to represent different underpinning constructs.

The proposed CSE methodology can complement traditional Rasch-model-based dimensionality analyses and provide additional nuances when seeking understanding of dimensionality aspects. The CSE methodology, however, is also an explanatory model in its own right, thereby providing support for validity. The set of explanatory variables tested here was similar for backward sequences as in our previous studies of forward sequences [6,38]. The strength of correlations between the quasi-theoretical measures of memory task difficulty against the empirical values were very similar for all eight combinations. One exception was when using only the digit recalling sequences (Equation (6)), where the correlation was lower. This might be related to previous work showing that the DST backward sequences are more challenging than the DST forward sequences, while task difficulty was similar in backward and forward recalling sequences of the CBT [17]. In turn, this suggests that additional variables may be needed to better explain forward and backward sequences and visuo-spatial and verbal memory recalling tasks.

Being able to explain the variation in task difficulties has been emphasized as a means of ensuring validity in measurements of person ability [33,34,35,36]. We argue that CSEs for task difficulty and person ability are both of significance for the validity in memory measurements [38]. This is especially true when one seeks to understand the underlying cognitive processes, as in the case of forward and backward sequences and visuo-spatial and verbal memory. Thus, a next proposed step to further understand (or to provide additional viewpoints) would be to explore if the same set of explanatory variables can be applied to explain different measures of person ability. Examples of explanatory variables for measures of person ability could be brain volumes and biomarkers [38,53] or the person’s ability to pay attention to the task to be performed using his or her executive functions, but an opening in line with our entropy argument would be to include functional brain networks and measures of connectivity [6]. Such an extension would also benefit from including other external structures in the other components of the measurement system (i.e., *Environment*, *Method,* and *Operator*).

Furthermore, as pointed out above, most analyses of memory tests have, to date, unfortunately not accounted for ordinality nor addressed separability between item and person attributes. This includes some of the recent research on forward and backward sequences and visuo-spatial and verbal memory tests [13,14,17,18,21,22,24]. Not properly separating item and person attributes hinders understanding of the cognitive processes underlying different cognitive domains. Thus, we would stress the importance of not only considering our CSE methodology, but also the significance of properly applying a measurement system approach where the human responder acts as instrument [30] and exploiting the unique properties of the Rasch model [37] in future studies of visuo-spatial and phonological processes in backward sequences (as suggested in [21]).

It was evident that, for the combined CSEs (Equations (5)–(8)), the measurement uncertainties in the β-coefficients were smaller compared to the individual CSEs (Equations (1)–(4)). This is not surprising, as they include more items, which are known to lead to reduced measurement uncertainties [38]. It is, however, worth noting that the CSEs, including backward recalling sequences, showed larger measurement uncertainties compared to corresponding CSEs for forward recalling sequences. An explanation of this could be larger measurement uncertainties for the empirical task difficulties for some of the analyses [38], such as the backward sequences. This was only true for the easiest item in the CBT, which reasonably should have the largest impact. Another explanation could be the variation in explanatory variables owing to the test design, such as differences in reversals and average distances, and may warrant adding sequences with the same entropy but with more variation in the other explanatory variables.

The NeuroMET cohort was designed to include subjects having a wide range of ability—from the least able to the most able. This is well-established practice in psychometrics to ensure maximum variation and, in turn, to make sure that subjects with different abilities can be measured on the same “ruler”. However, when different groups of subjects—such as the less cognitively able (e.g., patients with AD and MCI) or the more cognitively able (e.g., HC or persons with SCD)—perform differently on different kind of items, this can make it challenging to achieve metrological invariance. For instance, Muangpaisan et al. [54] proposed that backward recall better predicts MCI than forward recall. Serial position effects (SPEs) in word learning lists can also be a marker of AD and MCI [55]. Our previous work [8] on the diagnostic potential of SPEs in AVLT investigated a potential breakdown in the assumption of specific objectivity of the RMT, which is needed for metrological invariance. Understanding the limits to unidimensionality was achieved by successfully explaining PCA loading in both CSE formulation and in logistics regression residuals in terms of entropy, including SPEs using the Brillouin [40] formula in word learning list tests. Our work [8] indeed showed some effects of SPE scale distortion that might be correlated with different diagnostic groups, although measurement uncertainties were relatively large (reflecting the limited sample size). We suggested, nevertheless, that [8] *what appears to be the case is that, over and above individual variations in a person’s ability, there is an overall shift in the person’s ability for each clinical group. Whether one regards that as a change in ability or a change in task difficulty is a moot point.*

To conclude, our previously presented methodology—CSE—for testing theory of constructs and the role of entropy in explaining memory task difficulty [6,8] has successfully been replicated for backward sequences. Among the three explanatory variables studied (*entropy*, *reversals,* and *average distance*), *entropy* was the dominating term in explaining task difficulty. Moreover, analyses of dimensionality and the larger measurement uncertainties in the CSEs for the backward sequences suggest caution when unifying a single unidimensional construct based on forward and backward sequences with visuo-spatial and verbal memory tasks when measuring a person’s memory ability.

## Figures and Tables

**Table 1 entropy-25-00813-t001:** Dimensionality analyses.

Title 1	CBT		DST		Forward Sequences		Backward Sequences	
	Individual Sequences ^a^	Subtests ^b^	Individual Sequences ^a^	Subtests ^b^	Individual Sequences ^c^	Subtests ^d^	Individual Sequences ^e^	Subtests ^f^
PCA:								
PC1 eigenv	1.88		2.12		2.34		2.65	
PC2 eigenv	1.70		1.54		1.62		1.84	
Ratio ^g^	1.11		1.38		1.44		1.44	
Subtest analysis:								
α	0.79	0.73	0.83	0.76	0.76	0.53	0.80	0.59
α diff. ^h^		−0.06		−0.07		−0.23		−0.21
*c* ^i^		0.42		0.46		0.93		0.87
*r* ^j^		0.85		0.83		0.54		0.57
*A* ^k^		0.92		0.91		0.71		0.84
*t*-tests:								
*p* < 0.05 ^l^	8.43	4.24	13.55	3.94	20.54	6.71	19.09	3.34
95% CI ^m^	5.86–11.96	2.48–7.05	10.26–17.68	2.25–6.68	16.53–25.23	4.43–9.99	15.20–23.69	1.81–5.96

^a^ Data analysis based on individual forward and backward sequences combined. ^b^ Data analysis based on two subtests: one comprising forward sequences and one comprising backward sequences. ^c^ Data analysis based on individual forward CBT and DST sequences combined. ^d^ Data analysis based on two subtests: one comprising forward CBT sequences and one comprising forward DST sequences. ^e^ Data analysis based on individual backward CBT and DST sequences combined. ^f^ Data analysis based on two subtests: one comprising backward CBT sequences and one comprising backward DST sequences. ^g^ Ratio between PC1/PC2 eigenvalues. ^h^ Difference between α from analysis based on individual sequences and subtests. ^i^ Variance that is unique to the subtests. ^j^ Latent correlation between subtests corrected for attenuation due to measurement errors. ^k^ Non-error variance common to subtests. ^l^ Proportion of persons in the sample with significantly (*p* < 0.05) different person locations as estimated from two subsets of items. ^m^ 95% binomial Agresti–Coull confidence interval of the proportion of persons in the sample with significantly (*p* < 0.05) different person locations as estimated from two subsets of items. CBT, Corsi Block Test; DST, Digit Span Test; PCA, principal component analysis; PC1, first principal component; PC2, second principal component; eigenv, eigenvalue; α, coefficient alpha; diff., difference.

## Data Availability

Data used can be downloaded at https://zenodo.org/record/7439708#.Y5qgh3bMJPY.

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
