# Peer review of "Forward and Backward Recalling Sequences in Spatial and Verbal Memory Tasks: What Do We Measure?"

_entropy, 2023, doi:10.3390/e25050813_

Round 1
Reviewer 1 Report
The paper deals about the problem of modelling how visuo-spatial and verbal digits working memory in backward recalling tasks may function, also in relation with forward recalling tasks when a person should remember a block sequence (like in Corsi Block Test) or a digit sequence (like in Digit Span Test). The paper is part of the NeuroMet project, and the main topics treated are cognitive abilities or the problem of finding a cognitive model to explain and measure the task difficulty and the person ability in executing tasks of the above kind.
In particular, the paper introduces a combination of well known and new metrics for measuring the impact of backward sequences on visuo-spatial and verbal memory tasks, respectively. The method relies on three main points: humans act as instruments; the exploitation of the IRT, Rasch model; provide construct specification equations (CSE) for validation. The latter is used in particular to spot explanatory models for working memory task difficulty, together with the entropy metric, measuring high order (low entropy) and hence less difficulty for one task, reversals and average distance. This approach is applied for backward recalling block tapping and digit sequences, to provide a "better understanding of what is being measured with CBT and DST backward".
The study is well described, and all the information regarding the experiment design, the methods applied and the resulty analysis and discussion seem clear and robust. The main conclusions are based on the initial hypothesis/setting:
1) explaining memory task difficulty in backward recalling on the same set of explanatory variables as for the forward sequences
2) compare memory task difficulty between the visuo-spatial- and verbal memory tests
3) compare Rash models with CSE in these kind of studies
The results derived from eight variants of CSE seem to be that:
1) entropy is the dominating explanatory variable, althogh all the three variables (entropy, reversals and average distance) have a high measurement uncertainty based of beta-coefficients values, the strongest one including a combination of items from CBT and DST backward and forward sequences
2) there is some evidence of multidimensionality when comparing forward vs. backward sequences in some combinations of tests
3) The proposed CSE methodology can complement traditional Rasch-model based dimensionality analyses and provide additional nuances, when seeking understanding of dimensionality aspects
The paper concludes by claiming that new explanatory variables are necessary to measure person ability, e.g., neurophysiological signals.
As said above, I found the paper clear and fair, moderately optimist in discussing the results obtained, but able to add a new piece of knowledge to previous studies, by demonstrating the meaning of entropy in explaining task difficulty, for example, and the importance of providing CSE for supporting Rash in finding more nuanced results related to measurement constructs. I have only a little complaint related to the fact that the paper mentioned the possibility to measure person ability with biomarkers, but no additional variables were added in this direction.
However, I would propose acceptance in the journal, without any further improvements.
I only recommend authors to read the paper once more, as there are few typos to amend here and there in the manuscript.
Author Response
Reviewer 1, first of all, we would like to thank you for your appreciating words and nice summary of our paper. Below we have responded to your two comments.
|
Reviewer comment |
Response |
|
I have only a little complaint related to the fact that the paper mentioned the possibility to measure person ability with biomarkers, but no additional variables were added in this direction. |
We have included a sub-clause in lines 355-356 and a sentence in lines 358-360. |
|
I only recommend authors to read the paper once more, as there are few typos to amend here and there in the manuscript. |
Manuscript re-read and revised where necessary – see tracked changes. |
Reviewer 2 Report
Authors have reported that distance in cm for CBT, while they did not report what is the distance between digits in DST (line 161). If these tasks are recall tasks, I find it difficult to understand how these distances are in cm. Authors have not described what 'reversals' are in their equations.
Is the entropy measured in task structure? If yes, authors should explain how does a measurement in external structure affect the ability of the brain to successfully engage with the task.
Authors should clearly define 'difficult' task in physiological terms. Does 'a difficult' task have a higher failure rate? They must also discuss how an ordered task is less difficult for the brain interacting with task-based stimuli.
Author Response
Reviewer 2, below we have responded to your comments.
|
Reviewer comment |
Response |
|
Authors have reported that distance in cm for CBT, while they did not report what is the distance between digits in DST (line 161). If these tasks are recall tasks, I find it difficult to understand how these distances are in cm. |
Text at lines 166- 8 revised.
|
|
Authors have not described what 'reversals' are in their equations. |
Text at lines 163 – 6 revised. |
|
Is the entropy measured in task structure? If yes, authors should explain how does a measurement in external structure affect the ability of the brain to successfully engage with the task. |
We have modified the text, for instance at lines 166 ff, by clarifying the principle of specific objectivity which, according to Rasch, allows separate estimates of task difficulty and person ability. Our theories predict task difficulty based solely on sequence structure as a kind of “average” across the cohort, and independent of the particular ability of any individual cohort member, according to this principle.
The principle of specific objectivity has, of course, to be verified. The paragraph starting in line 447 already indicates our intended next steps to study brain activity related to memory ability where such a verification could be performed. We have extended that paragraph to indicate that intention. |
|
Authors should clearly define 'difficult' task in physiological terms. Does 'a difficult' task have a higher failure rate? They must also discuss how an ordered task is less difficult for the brain interacting with task-based stimuli. |
The Rasch model is a probabilistic model and we have added new text, both in the Introduction (see preceding point) and in section 2.2, about its underpinnings for person ability and task difficulty. |